# Estimating Convergence of Markov chains with $L$-Lag Couplings

**Niloy Biswas**
Harvard University
niloy_biswas@g.harvard.edu

**Pierre E. Jacob**
Harvard University
pjacob@fas.harvard.edu

**Paul Vanetti**
University of Oxford
paul.vanetti@spc.ox.ac.uk

## Abstract

Markov chain Monte Carlo (MCMC) methods generate samples that are asymptotically distributed from a target distribution of interest as the number of iterations goes to infinity. Various theoretical results provide upper bounds on the distance between the target and marginal distribution after a fixed number of iterations. These upper bounds are on a case by case basis and typically involve intractable quantities, which limits their use for practitioners. We introduce $L$-lag couplings to generate computable, non-asymptotic upper bound estimates for the total variation or the Wasserstein distance of general Markov chains. We apply $L$-lag couplings to the tasks of (i) determining MCMC burn-in, (ii) comparing different MCMC algorithms with the same target, and (iii) comparing exact and approximate MCMC. Lastly, we (iv) assess the bias of sequential Monte Carlo and self-normalized importance samplers.

## 1 Introduction

Markov chain Monte Carlo (MCMC) algorithms generate Markov chains that are invariant with respect to probability distributions that we wish to approximate. Numerous works help understanding the convergence of these chains to their invariant distributions, hereafter denoted by $\pi$. Denote by $\pi_t$ the marginal distribution of the chain $(X_t)_{t \geq 0}$ at time $t$. The discrepancy between $\pi_t$ and $\pi$ can be measured in different ways, typically the total variation (TV) distance or the Wasserstein distance in the MCMC literature. Various results provide upper bounds on this distance, of the form $C(\pi_0)f(t)$, where $C(\pi_0) < \infty$ depends on $\pi_0$ but not on $t$, and where $f(t)$ decreases to zero as $t$ goes to infinity, typically geometrically; see Section 3 in [48] for a gentle survey, and [17, 13, 18] for recent examples. These results typically relate convergence rates to the dimension of the state space or to various features of the target. Often these results do not provide computable bounds on the distance between $\pi_t$ and $\pi$, as $C(\pi_0)$ and $f(t)$ typically feature unknown constants; although see [49] where these constants can be bounded analytically, and [12] for examples where they can be numerically approximated.

Various tools have been developed to assess the quality of MCMC estimates. Some focus on the behaviour of the chains assuming stationarity, comparing averages computed within and across chains, or defining various notions of effective sample sizes based on asymptotic variance estimates (e.g. [20, 21, 19, 56], [46, Chapter 8]). Few tools provide computable bounds on the distance between $\pi_t$ and $\pi$ for a fixed $t$; some are mentioned in [6] for Gibbs samplers with tractable transition kernels. Notable exceptions, beyond [12] mentioned above, include the method of [31, 32] which relies on coupled Markov chains. A comparison with our proposed method will be given in Section 2.4.

We propose to use $L$-lag couplings of Markov chains to estimate the distance between $\pi_t$ and $\pi$ for a fixed time $t$, building on 1-lag couplings used to obtain unbiased estimators in [23, 29]. The discussion of [29] mentions that upper bounds on the TV between $\pi_t$ and $\pi$ can be estimated with such couplings. We generalize this idea to $L$-lag couplings, which provide sharper bounds, particularly

for small values of $t$. The proposed technique extends to a class of probability metrics [52] beyond TV. We demonstrate numerically that the bounds provide a practical assessment of convergence for various popular MCMC algorithms, on either discrete or continuous and possibly high-dimensional spaces. The proposed bounds can be used to (i) determine burn-in period for MCMC estimates, to (ii) compare different MCMC algorithms targeting the same distribution, or to (iii) compare exact and approximate MCMC algorithms, such as Unadjusted and Metropolis-adjusted Langevin algorithms, providing a computational companion to studies such as [18]. We also (iv) assess the bias of sequential Monte Carlo and self-normalized importance samplers.

In Section 2 we introduce $L$-lag couplings to estimate metrics between marginal and invariant distributions of a Markov chain. We illustrate the method on simple examples, discuss the choice of $L$, and compare with the approach of [31]. In Section 3 we consider applications including Gibbs samplers on the Ising model and gradient-based MCMC algorithms on log-concave targets. In Section 4 we assess the bias of sequential Monte Carlo and self-normalized importance samplers. All scripts in R are available at `https://github.com/niloyb/LlagCouplings`.

## 2 $L$-lag couplings

Consider two Markov chains $(X_t)_{t\geq 0}$, $(Y_t)_{t\geq 0}$, each with the same initial distribution $\pi_0$ and Markov kernel $K$ on $(\mathbb{R}^d, \mathcal{B}(\mathbb{R}^d))$ which is $\pi$-invariant. Choose some integer $L \geq 1$ as the lag parameter. We generate the two chains using Algorithm 1. The joint Markov kernel $\bar{K}$ on $(\mathbb{R}^d \times \mathbb{R}^d, \mathcal{B}(\mathbb{R}^d \times \mathbb{R}^d))$ is such that, for all $x$, $y$, $\bar{K}((x,y),(\cdot,\mathbb{R}^d)) = K(x,\cdot)$, and $\bar{K}((x,y),(\mathbb{R}^d,\cdot)) = K(y,\cdot)$. This ensures that $X_t$ and $Y_t$ have the same marginal distribution at all times $t$. Furthermore, $\bar{K}$ is constructed such that the pair of chains can meet exactly after a random number of steps, i.e. the meeting time $\tau^{(L)} := \inf\{t > L : X_t = Y_{t-L}\}$ is almost surely finite. Finally we assume that the chains remain faithful after meeting, i.e. $X_t = Y_{t-L}$ for all $t \geq \tau^{(L)}$.

Various constructions for $\bar{K}$ have been derived in the literature: for instance coupled Metropolis-Hastings and Gibbs kernels in [31, 29], coupled Hamiltonian Monte Carlo kernels in [36, 5, 26], and coupled particle Gibbs samplers in [9, 3, 28].

---

**Algorithm 1:** Sampling $L$-lag meeting times

---

**Input:** lag $L \geq 1$, initial distribution $\pi_0$, single kernel $K$ and joint kernel $\bar{K}$
**Output:** meeting time $\tau^{(L)}$, and chains $(X_t)_{0\leq t\leq \tau^{(L)}}$, $(Y_t)_{0\leq t\leq \tau^{(L)}-L}$
Initialize: generate $X_0 \sim \pi_0$, $X_t|X_{t-1} \sim K(X_{t-1},\cdot)$ for $t = 1,\ldots,L$, and $Y_0 \sim \pi_0$
**for** $t > L$ **do**
    Sample $(X_t, Y_{t-L})|(X_{t-1}, Y_{t-L-1}) \sim \bar{K}((X_{t-1}, Y_{t-L-1}),\cdot)$
    **if** $X_t = Y_{t-L}$ **then** **return** $\tau^{(L)} := t$, and chains $(X_t)_{0\leq t\leq \tau^{(L)}}$, $(Y_t)_{0\leq t\leq \tau^{(L)}-L}$
**end**

---

We next introduce integral probability metrics (IPMs, e.g. [52]).

**Definition 2.1.** *(Integral Probability Metric). Let $\mathcal{H}$ be a class of real-valued functions on a measurable space $\mathcal{X}$. For all probability measures $P, Q$ on $\mathcal{X}$, the corresponding IPM is defined as:*

$$d_{\mathcal{H}}(P,Q) := \sup_{h\in\mathcal{H}} \left| \mathbb{E}_{X\sim P}[h(X)] - \mathbb{E}_{X\sim Q}[h(X)] \right|. \tag{1}$$

Common IPMs include total variation distance $d_{\mathrm{TV}}$ with $\mathcal{H} := \{h : \sup_{x\in\mathcal{X}} |h(x)| \leq 1/2\}$, and 1-Wasserstein distance $d_{\mathrm{W}}$ with $\mathcal{H} = \{h : |h(x) - h(y)| \leq d_{\mathcal{X}}(x,y), \forall x, y \in \mathcal{X}\}$, where $d_{\mathcal{X}}$ is a metric on $\mathcal{X}$ [42]. Our proposed method applies to IPMs such that $\sup_{h\in\mathcal{H}} |h(x) - h(y)| \leq M_{\mathcal{H}}(x,y)$ for all $x, y \in \mathcal{X}$, for some computable function $M_{\mathcal{H}}$ on $\mathcal{X} \times \mathcal{X}$. For $d_{\mathrm{TV}}$ we have $M_{\mathcal{H}}(x,y) = 1$, and for $d_{\mathrm{W}}$ we have $M_{\mathcal{H}}(x,y) = d_{\mathcal{X}}(x,y)$.

With a similar motivation for the assessment of sample approximations, and not restricted to the MCMC setting, [25] considers a restricted class of functions $\mathcal{H}$ to develop a specific measure of sample quality based on Stein's identity. [35, 10] combine Stein's identity with reproducing kernel Hilbert space theory to develop goodness-of-fit tests. [24] obtains further results and draws connections to the literature on couplings of Markov processes. Here we directly aim at upper bounds on the total variation and Wasserstein distance. The total variation controls the maximal difference

between the masses assigned by $\pi_t$ and $\pi$ on any measurable set, and thus directly helps assessing the error of histograms of the target marginals. The 1-Wasserstein distance controls the error made on expectations of 1-Lipschitz functions, which with $\mathcal{X} = \mathbb{R}^d$ and $d_\mathcal{X}(x, y) = \|x - y\|_1$ (the $L_1$ norm on $\mathbb{R}^d$) include all first moments.

## 2.1 Main results

We make the three following assumptions similar to those of [29].

**Assumption 2.2.** *(Marginal convergence and moments.)* *For all $h \in \mathcal{H}$, as $t \to \infty$, $\mathbb{E}[h(X_t)] \to \mathbb{E}_{X \sim \pi}[h(X)]$. Also, $\exists \eta > 0, D < \infty$ such that $\mathbb{E}[M_\mathcal{H}(X_t, Y_{t-L})^{2+\eta}] \leq D$ for all $t \geq L$.*

The above assumption is on the marginal convergence of the MCMC algorithm and on the moments of the associated chains. The next assumptions are on the coupling operated by the joint kernel $\bar{K}$.

**Assumption 2.3.** *(Sub-exponential tails of meeting times.)* *The chains are such that the meeting time $\tau^{(L)} := \inf\{t > L : X_t = Y_{t-L}\}$ satisfies $\mathbb{P}(\frac{\tau^{(L)} - L}{L} > t) \leq C\delta^t$ for all $t \geq 0$, for some constants $C < \infty$ and $\delta \in (0, 1)$.*

The above assumption can be relaxed to allow for polynomial tails as in [37]. The final assumption on faithfulness is typically satisfied by design.

**Assumption 2.4.** *(Faithfulness.)* *The chains stay together after meeting: $X_t = Y_{t-L}$ for all $t \geq \tau^{(L)}$.*

We assume that the three assumptions above hold in the rest of the article. The following theorem is our main result.

**Theorem 2.5.** *(Upper bounds.)* *For an IPM with function set $\mathcal{H}$ and upper bound $M_\mathcal{H}$, with the Markov chains $(X_t)_{t \geq 0}, (Y_t)_{t \geq 0}$ satisfying the above assumptions, for any $L \geq 1$, and any $t \geq 0$,*

$$d_\mathcal{H}(\pi_t, \pi) \leq \mathbb{E}\Big[ \sum_{j=1}^{\lceil \frac{\tau^{(L)} - L - t}{L} \rceil} M_\mathcal{H}(X_{t+jL}, Y_{t+(j-1)L}) \Big]. \tag{2}$$

Here $\lceil x \rceil$ denotes the smallest integer above $x$, for $x \in \mathbb{R}$. When $\lceil (\tau^{(L)} - L - t)/L \rceil \leq 0$, the sum in inequality (2) is set to zero by convention. We next give an informal proof. Seeing the invariant distribution $\pi$ as the limit of $\pi_t$ as $t \to \infty$, applying triangle inequalities, recalling that $d_\mathcal{H}(\pi_s, \pi_t) \leq \mathbb{E}[M_\mathcal{H}(X_s, X_t)]$ for all $s, t$, we obtain

$$d_\mathcal{H}(\pi_t, \pi) \leq \sum_{j=1}^{\infty} d_\mathcal{H}(\pi_{t+jL}, \pi_{t+(j-1)L}) \leq \sum_{j=1}^{\infty} \mathbb{E}[M_\mathcal{H}(X_{t+jL}, Y_{t+(j-1)L})]. \tag{3}$$

The right-hand side of (2) is retrieved by swapping expectation and limit, and noting that terms indexed by $j > \lceil (\tau^{(L)} - L - t)/L \rceil$ are equal to zero by Assumption 2.4. The above reasoning highlights that increasing $L$ leads to sharper bounds through the use of fewer triangle inequalities. An alternate, formal proof based on an unbiased estimation argument is given in supplementary material.

Theorem 2.5 gives the following bounds for $d_{\mathrm{TV}}$ and $d_\mathrm{W}$,

$$d_{\mathrm{TV}}(\pi_t, \pi) \leq \mathbb{E}\Big[ \max(0, \lceil \frac{\tau^{(L)} - L - t}{L} \rceil) \Big], \tag{4}$$

$$d_\mathrm{W}(\pi_t, \pi) \leq \mathbb{E}\Big[ \sum_{j=1}^{\lceil \frac{\tau^{(L)} - L - t}{L} \rceil} d_\mathcal{X}(X_{t+jL}, Y_{t+(j-1)L}) \Big]. \tag{5}$$

For the total variation distance, the boundedness part of Assumption 2.2 is directly satisfied. For the 1-Wasserstein distance on $\mathbb{R}^d$ with $d_\mathcal{X}(x, y) = \|x - y\|_1$ (the $L_1$ norm on $\mathbb{R}^d$), the boundedness part is equivalent to a uniform bound of $(2 + \eta)$-th moments of the marginal distributions for some $\eta > 0$.

We emphasize that the proposed bounds can be estimated directly by running Algorithm 1 $N$ times independently, and using empirical averages. All details of the MCMC algorithms and their couplings mentioned below are provided in the supplementary material.

## 2.2 Stylized examples

### 2.2.1 A univariate Normal

We consider a Normal example where we can compute total variation and 1-Wasserstein distances (using the $L_1$ norm on $\mathbb{R}$ throughout) exactly. The target $\pi$ is $\mathcal{N}(0,1)$ and the kernel $K$ is that of a Normal random walk Metropolis-Hastings (MH) with step size $\sigma_{\text{MH}} = 0.5$. We set the initial distribution $\pi_0$ to be a point mass at 10. The joint kernel $\bar{K}$ operates as follows. Given $(X_{t-1}, Y_{t-L-1})$, sample $(X^\star, Y^\star)$ from a maximal coupling of $p := \mathcal{N}(X_{t-1}, \sigma_{\text{MH}}^2)$ and $q := \mathcal{N}(Y_{t-L-1}, \sigma_{\text{MH}}^2)$. This is done using Algorithm 2, which ensures $X^\star \sim p$, $Y^\star \sim q$ and $\mathbb{P}(X^\star \neq Y^\star) = d_{\text{TV}}(p, q)$.

---

**Algorithm 2:** A maximal coupling of $p$ and $q$

---

Sample $X^* \sim p$, and $W \sim \mathcal{U}(0,1)$
**if** $p(X^*)W \leq q(X^*)$ **then** set $Y^* = X^*$ and **return** $(X^*, Y^*)$
**else** sample $\tilde{Y} \sim q$ and $\tilde{W} \sim \mathcal{U}(0,1)$ until $q(\tilde{Y})\tilde{W} > p(\tilde{Y})$. Set $Y^* = \tilde{Y}$ and **return** $(X^*, Y^*)$

---

Having obtained $(X^\star, Y^\star)$, sample $U \sim \mathcal{U}(0,1)$; set $X_t = X^\star$ if $U < \pi(X^\star)/\pi(X_{t-1})$; otherwise set $X_t = X_{t-1}$. With the same $U$, set $Y_{t-L} = Y^\star$ if $U < \pi(Y^\star)/\pi(Y_{t-L-1})$; otherwise set $Y_{t-L} = Y_{t-L-1}$. Such a kernel $\bar{K}$ is a coupling of $K$ with itself, and Assumption 2.4 holds by design. The verification of Assumption 2.3 is harder but can be done via drift conditions in various cases; we refer to [29] for more discussion.

Figure 1 shows the evolution of the marginal distribution of the chain, and the TV and 1-Wasserstein distance upper bounds. We use $L = 1$ and $L = 150$. For each $L$, $N = 10000$ independent runs of Algorithm 1 were performed to estimate the bounds in Theorem 2.5 by empirical averages. Exact distances are shown for comparison. Tighter bounds are obtained with larger values of $L$, as discussed further in Section 2.3.

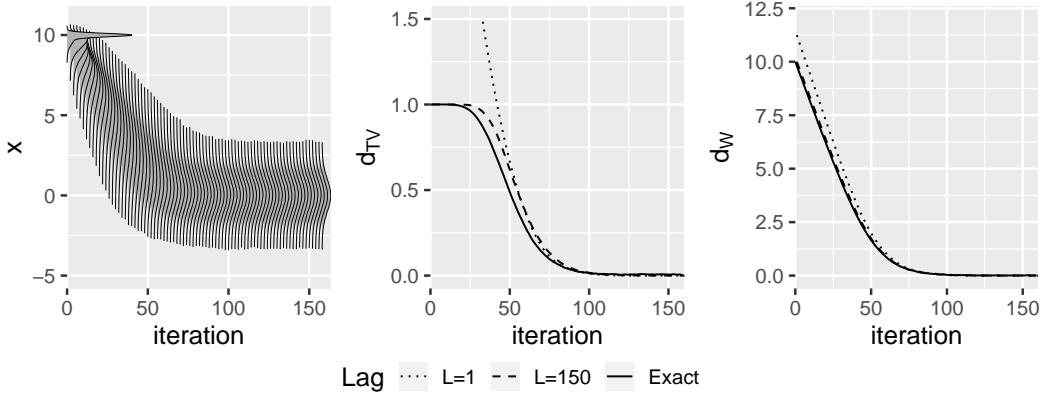

Figure 1: Marginal distributions of the chain (left), and upper bounds on the total variation (middle) and the 1-Wasserstein distance (right) between $\pi_t$ and $\pi$, for a Metropolis-Hastings algorithm targeting $\mathcal{N}(0,1)$ and starting from a Dirac mass at 10. With $L = 150$ the estimated upper bounds for both are close to the exact distances.

### 2.2.2 A bimodal target

We consider a bimodal target to illustrate the limitations of the proposed technique. The target is $\pi = \frac{1}{2}\mathcal{N}(-4,1) + \frac{1}{2}\mathcal{N}(4,1)$, as in Section 5.1 of [29]. The MCMC algorithm is again random walk MH, with $\sigma_{\text{MH}} = 1$, $\pi_0 = \mathcal{N}(10,1)$. Now, the chains struggle to jump between the modes, as seen in Figure 2 (left), which shows a histogram of the 500th marginal distribution from 1000 independent chains. Figure 2 (right) shows the TV upper bound estimates for lags $L = 1$ and $L = 18000$ (considered very large), obtained with $N \in \{1000, 5000, 10000\}$ independent runs of Algorithm 1.

With $L = 18000$, we do not see a difference between the obtained upper bounds, which suggests that the variance of the estimators is small for the different values of $N$. In contrast, the dashed line bounds corresponding to lag $L = 1$ are very different. This is because, over 1000 experiments,

the 1-lag meetings always occurred quickly in the mode nearest to the initial distribution. However, over 5000 and 10000 experiments, there were instances where one of the two chains jumped to the other mode before meeting, resulting in a much longer meeting time. Thus the results obtained with $N = 1000$ repeats can be misleading. This is a manifestation of the estimation error associated with empirical averages, which are not guaranteed to be accurate after any fixed number $N$ of repeats. The shape of the bounds obtained with $L = 18000$, with a plateau, reflects how the chains first visit one of the modes, and then both.

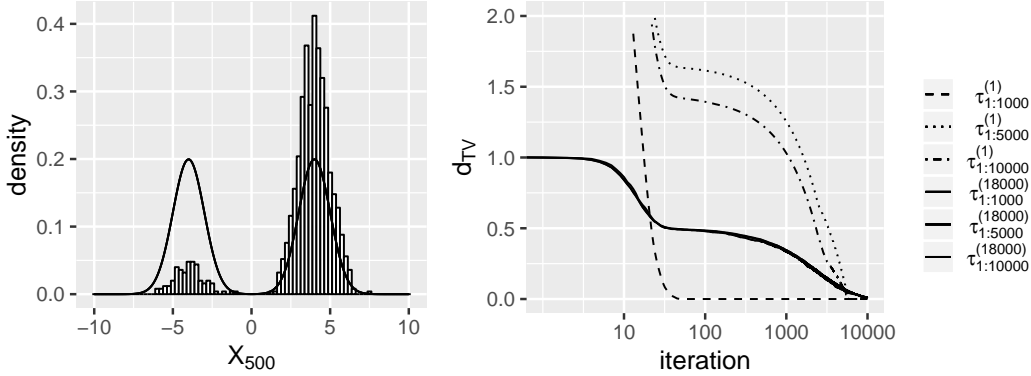

Figure 2: Metropolis-Hastings algorithm with $\pi_0 \sim \mathcal{N}(10, 1), \sigma_{\mathrm{MH}} = 1$ on a bimodal target. Left: Histogram of the 500th marginal distribution from 1000 independent chains, and target density in full line. Right: Total variation bounds obtained with lags $L \in \{1, 18000\}$ and $N \in \{1000, 5000, 10000\}$ independent runs of Algorithm 1.

## 2.3 Choice of lag $L$

Section 2.2.2 illustrates the importance of the choice of lag $L$. Obtaining $\tau^{(L)}$ requires sampling $L$ times from $K$ and $\tau^{(L)} - L$ from $\bar{K}$. When $L$ gets large, we can consider $X_L$ to be at stationarity, while $Y_0$ still follows $\pi_0$. Then the distribution of $\tau^{(L)} - L$ depends entirely on $\bar{K}$ and not on $L$. In that regime the cost of obtaining $\tau^{(L)}$ increases linearly in $L$. On the other hand, if $L$ is small, the cost might be dominated by the $\tau^{(L)} - L$ draws from $\bar{K}$. Thus increasing $L$ might not significantly impact the cost until the distribution of $\tau^{(L)} - L$ becomes stable in $L$.

The point of increasing $L$ is to obtain sharper bounds. For example, from (4) we see that, for fixed $t$, the variable in the expectation takes values in $[0, 1]$ with increasing probability as $L \to \infty$, resulting in upper bounds more likely to be in $[0, 1]$ and thus non-vacuous. The upper bound is also decreasing in $t$. This motivates the strategy of starting with $L = 1$, plotting the bounds as in Figure 1, and increasing $L$ until the estimated upper bound for $d_{\mathrm{TV}}(\pi_0, \pi)$ is close to 1.

Irrespective of the cost, the benefits of increasing $L$ eventually diminish: the upper bounds are loose to some extent since the coupling operated by $\bar{K}$ is not optimal [54]. The couplings considered in this work are chosen to be widely applicable but are not optimal in any way.

## 2.4 Comparison with Johnson's diagnostics

The proposed approach is similar to that proposed by Valen Johnson in [31], which works as follows. A number $c \geq 2$ of chains start from $\pi_0$ and evolve jointly (without time lags), such that they all coincide exactly after a random number of steps $T_c$, while each chain marginally evolves according to $K$. If we assume that any draw from $\pi_0$ would be accepted as a draw from $\pi$ in a rejection sampler with probability $1 - r$, then the main result of [31] provides the bound: $d_{\mathrm{TV}}(\pi_t, \pi) \leq \mathbb{P}(T_c > t) \times (1 - r^c)^{-1}$. As $c$ increases, for any $r \in (0, 1)$ the upper bound approaches $\mathbb{P}(T_c > t)$, which itself is small if $t$ is a large quantile of the meeting time $T_c$. A limitation of this result is its reliance on the quantity $r$, which might be unknown or very close to one in challenging settings. Another difference is that we rely on pairs of lagged chains and tune the lag $L$, while the tuning parameter in [31] is the number of coupled chains $c$.

## 3 Experiments and applications

### 3.1 Ising model

We consider an Ising model, where the target is defined on a large discrete space, namely a square lattice with $32 \times 32$ sites (each site has 4 neighbors) and periodic boundaries. For a state $x \in \{-1, +1\}^{32 \times 32}$, we define the target probability $\pi_\beta(x) \propto \exp(\beta \sum_{i \sim j} x_i x_j)$, where the sum is over all pairs $i$, $j$ of neighboring sites. As $\beta$ increases, the correlation between nearby sites increases and single-site Gibbs samplers are known to perform poorly [39]. Difficulties in the assessment of the convergence of these samplers are in part due to the discrete nature of the state space, which limits the possibilities of visual diagnostics. Users might observe trace plots of one-dimensional statistics of the chains, such as $x \mapsto \sum_{i \sim j} x_i x_j$, and declare convergence when the statistic seems to stabilize; see [55, 60] where trace plots of summary statistics are used to monitor Markov chains.

Here we compute the proposed upper bounds for the TV distance for two algorithms: a single site Gibbs sampler (SSG) and a parallel tempering (PT) algorithm, where different chains target different $\pi_\beta$ with SSG updates, and regularly attempt to swap their states [22, 53]. The initial distribution assigns $-1$ and $+1$ with equal probability on each site independently. For $\beta = 0.46$, we obtain TV bounds for SSG using a lag $L = 10^6$, and $N = 500$ independent repeats. For PT we use 12 chains, each targeting $\pi_\beta$ with $\beta$ in an equispaced grid ranging from 0.3 to 0.46, a frequency of swap moves of 0.02, and a lag $L = 2 \times 10^4$. The results are in Figure 3, where we see a plateau for the TV bounds on SSG and faster convergence for the TV bounds on PT. Our results are consistent with theoretical work on faster mixing times of PT targeting multimodal distributions including Ising models [59]. Note that the targets are different for both algorithms, as PT operates on an extended space. The behavior of meeting times of coupled chains motivated by the "coupling from the past" algorithm [44] for Ising models has been studied e.g. in [11].

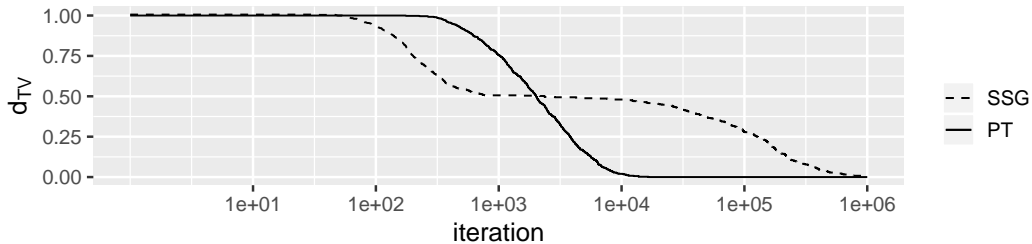

Figure 3: Single-site Gibbs (SSG) versus Parallel Tempering (PT) for an Ising model; bounds on the total variation distance between $\pi_t$ and $\pi$, for $t$ up to $10^6$ and inverse temperature $\beta = 0.46$.

### 3.2 Logistic regression

We next consider a target on a continuous state space defined as the posterior in a Bayesian logistic regression. Consider the German Credit data from [34]. There are $n = 1000$ binary responses $(Y_i)_{i=1}^n \in \{-1, 1\}^n$ indicating whether individuals are creditworthy or not creditworthy, and $d = 49$ covariates $x_i \in \mathbb{R}^d$ for each individual $i$. The logistic regression model states $\mathbb{P}(Y_i = y_i | x_i) = (1 + e^{-y_i x_i^T \beta})^{-1}$ with a normal prior $\beta \sim \mathcal{N}(0, 10 I_d)$. We can sample from the posterior using Hamiltonian Monte Carlo (HMC, [40]) or the Pólya-Gamma Gibbs sampler (PG, [43]). The former involves tuning parameters $\epsilon_{\text{HMC}}$ and $S_{\text{HMC}}$ corresponding to a step size and a number of steps in a leapfrog integration scheme performed at every iteration. We can use the proposed bounds to compare convergence associated with HMC for different $\epsilon_{\text{HMC}}, S_{\text{HMC}}$, and with the PG sampler. Figure 4 shows the total variation bounds for HMC with $\epsilon_{\text{HMC}} = 0.025$ and $S_{\text{HMC}} = 4, 5, 6, 7$ and the corresponding bound for the parameter-free PG sampler, both starting from $\pi_0 \sim \mathcal{N}(0, 10 I_d)$. In this example, the bounds are smaller for the PG sampler than for all HMC samplers under consideration.

We emphasize that the HMC tuning parameters associated with the fastest convergence to stationarity might not necessarily be optimal in terms of asymptotic variance of ergodic averages of functions of interest; see related discussions in [26]. Also, since the proposed upper bounds are not tight, the

true convergence rates of the Markov chains under consideration may be ordered differently. The proposed upper bounds still allow a comparison of how confident we can be about the bias of different MCMC algorithms after a fixed number of iterations.

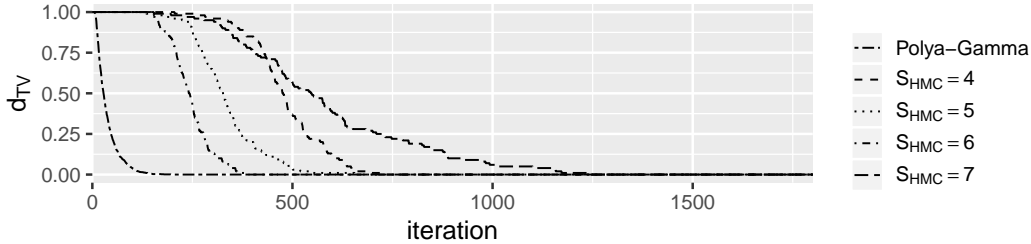

Figure 4: Proposed upper bounds on $d_{\text{TV}}(\pi_t, \pi)$ for a Pólya-Gamma Gibbs sampler and for Hamiltonian Monte Carlo on a 49-dimensional posterior distribution in a logistic regression model. For HMC the step size is $\epsilon_{\text{HMC}} = 0.025$ and the number of steps is $S_{\text{HMC}} = 4, 5, 6, 7$.

### 3.3 Comparison of exact and approximate MCMC algorithms

In various settings approximate MCMC methods trade off asymptotic unbiasedness for gains in computational speed, e.g. [30, 50, 14]. We compare an approximate MCMC method (Unadjusted Langevin Algorithm, ULA) with its exact counterpart (Metropolis-Adjusted Langevin Algorithm, MALA) in various dimensions. Our target is a multivariate normal:

$$\pi = \mathcal{N}(0, \Sigma) \text{ where } [\Sigma]_{i,j} = 0.5^{|i-j|} \text{ for } 1 \le i, j \le d.$$

Both MALA and ULA chains start from $\pi_0 \sim \mathcal{N}(0, I_d)$, and have step sizes of $d^{-1/6}$ and $0.1d^{-1/6}$ respectively. Step sizes are linked to an optimal result of [47], and the 0.1 multiplicative factor for ULA ensures that the target distribution for ULA is close to $\pi$ (see [13]). We can use couplings to study the mixing times $t_{\text{mix}}(\epsilon)$ of the two algorithms, where $t_{\text{mix}}(\epsilon) := \inf\{k \ge 0 : d_{\text{TV}}(\pi_k, \pi) < \epsilon\}$. Figure 5 highlights how the dimension impacts the estimated upper bounds on the mixing time $t_{\text{mix}}(0.25)$, calculated as $\inf\{k \ge 0 : \widehat{\mathbb{E}}[\max(0, \lceil (\tau^{(L)} - L - k)/L \rceil)] < 0.25\}$ where $\widehat{\mathbb{E}}$ denotes empirical averages. The results are consistent with the theoretical analysis in [18]. For a strongly log-concave target such as $\mathcal{N}(0, \Sigma)$, Table 2 of [18] indicates mixing time upper bounds of order $\mathcal{O}(d)$ and $\mathcal{O}(d^2)$ for ULA and MALA respectively (with a *non-warm* start centered at the unique mode of the target). In comparison to theoretical studies in [13, 18], our bounds can be directly estimated by simulation. On the other hand, the bounds in [13, 18] are more explicit about the impact of different aspects of the problem including dimension, step size, and features of the target.

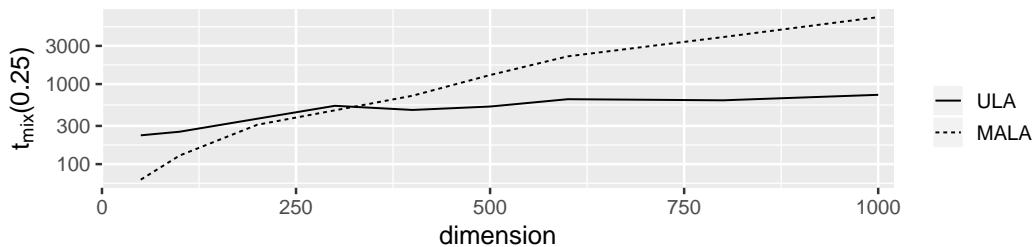

Figure 5: Mixing time bounds for ULA and MALA targeting a multivariate Normal distribution, as a function of the dimension. Mixing time $t_{\text{mix}}(0.25)$ denotes the first iteration $t$ for which the estimated TV between $\pi_t$ and $\pi$ is less than 0.25.

## 4 Assessing the bias of sequential Monte Carlo samplers

Lastly, we consider the bias associated with samples generated by sequential Monte Carlo (SMC) samplers [16]; the bias of self-normalized importance samplers can be treated similarly. Let $(w^n, \xi^n)_{n=1}^N$

be the weighted sample from an SMC sampler with $N$ particles targeting $\pi$, and let $q^{(N)}$ be the marginal distribution of a particle $\xi$ sampled among $(\xi^n)_{n=1}^N$ with probabilities $(w^n)_{n=1}^N$. Our aim is to upper bound a distance between $q^{(N)}$ and $\pi$ for a fixed $N$. We denote by $\hat{Z}$ the normalizing constant estimator generated by the SMC sampler.

The particle independent MH algorithm (PIMH, [2]) operates as an independent MH algorithm using SMC samplers as proposals. Let $(\hat{Z}_t)_{t\geq 0}$ be the normalizing constant estimates from a PIMH chain. Consider an $L$-lag coupling of a pair of such PIMH chains as introduced in [38], initializing the chains by running an SMC sampler. Here $\tau^{(L)}$ is constructed so that it can be equal to $L$ with positive probability; more precisely,

$$\tau^{(L)} - (L-1)\big|\hat{Z}_{L-1} \sim \mathrm{Geometric}(\alpha(\hat{Z}_{L-1})), \tag{6}$$

where $\alpha(\hat{Z}) := \mathbb{E}\big[\min(1, \hat{Z}^*/\hat{Z})\big|\hat{Z}\big]$ is the average acceptance probability of PIMH, from a state with normalizing constant estimate $\hat{Z}$; see [38, Proposition 8] for a formal statement in the case of 1-lag couplings. With this insight, we can bound the TV distance between the target and particles generated by SMC samplers, using Theorem 2.5 applied with $t = 0$. Details are in supplementary material. We obtain

$$d_{\mathrm{TV}}(q^{(N)}, \pi) \leq \mathbb{E}\Big[\max(0, \lceil \frac{\tau^{(L)} - L}{L}\rceil)\Big] = \mathbb{E}\Big[\frac{1 - \alpha(\hat{Z}_{L-1})}{1 - (1 - \alpha(\hat{Z}_{L-1}))^L}\Big]. \tag{7}$$

The bound in (7) depends only on the distribution of the normalizing constant estimator $\hat{Z}$, and can be estimated using independent runs of the SMC sampler. We can also estimate the distribution of $\hat{Z}$ from a single SMC sampler by appealing to large asymptotic results such as [4], combined with asymptotically valid variance estimators such as [33]. As $N$ goes to infinity we expect $\alpha(\hat{Z}_{L-1})$ to approach one and the proposed upper bound to go to zero. The proposed bound aligns with the common practice of considering the variance of $\hat{Z}$ as a measure of global performance of SMC samplers.

Existing TV bounds for particle approximations, such as those in [15, Chapter 8] and [27], are more informative qualitatively but harder to approximate numerically. The result also applies to self-normalized importance samplers (see [46, Chapter 3] and [41, Chapter 8]). In that case [1, Theorem 2.1] shows $d_{\mathrm{TV}}(q^{(N)}, \pi) \leq 6N^{-1}\rho$ for $\rho = \mathbb{E}_{\xi \sim q}[w(\xi)^2]/\mathbb{E}_{\xi \sim q}[w(\xi)]^2$, with $w$ the importance sampling weight function, which is a simpler and more informative bound; see also [8] for related results and concentration inequalities.

## 5 Discussion

The proposed method can be used to obtain guidance on the choice of burn-in, to compare different MCMC algorithms targeting the same distribution, and to compare mixing times of approximate and exact MCMC methods. The main requirement for the application of the method is the ability to generate coupled Markov chains that can meet exactly after a random but finite number of iterations. The couplings employed here, and described in supplementary materials, are not optimal in any way. As the couplings are algorithm-specific and not target-specific, they can potentially be added to statistical software such as PyMC3 [51] or Stan [7].

The bounds are not tight, in part due to the couplings not being maximal [54], but experiments suggest that they can be practical. The proposed bounds go to zero as $t$ increases, making them informative at least for large enough $t$. The combination of time lags and coupling of more than two chains as in [31] could lead to new diagnostics. Further research might also complement the proposed upper bounds with lower bounds, obtained by considering specific functions among the classes of functions used to define the integral probability metrics.

**Acknowledgments.** The authors are grateful to Espen Bernton, Nicolas Chopin, Andrew Gelman, Lester Mackey, John O'Leary, Christian Robert, Jeffrey Rosenthal, James Scott, Aki Vehtari and reviewers for helpful comments on an earlier version of the manuscript. The second author gratefully acknowledges support by the National Science Foundation through awards DMS-1712872 and DMS-1844695. The figures were created with packages [58, 57] in R Core Team [45].

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
