[Supplementary Material]

# Estimating Convergence of Markov chains with $L$-Lag Couplings: Supplementary material

**Niloy Biswas**
Harvard University
niloy_biswas@g.harvard.edu

**Pierre E. Jacob**
Harvard University
pjacob@fas.harvard.edu

**Paul Vanetti**
University of Oxford
paul.vanetti@spc.ox.ac.uk

## 1 Proofs

### 1.1 $L$-lag unbiased estimators

Our motivation for Theorem 2.5 comes from recent works on unbiased MCMC estimators using couplings [7, 5]. In particular, extending the unbiased estimator from [7] that corresponds to a lag $L = 1$, we first construct the $L$-lag estimator with an arbitrary $L \geq 1$ as

$$H_t^{(L)}(X,Y) := h(X_t) + \sum_{j=1}^{\left\lceil \frac{\tau^{(L)} - L - t}{L} \right\rceil} h(X_{t+jL}) - h(Y_{t+(j-1)L}). \tag{1}$$

where $h \in \mathcal{H}$, chains $(X_t)_{t \geq 0}$, $(Y_t)_{t \geq 0}$ marginally have the same initial distribution $\pi_0$ and Markov transition kernel $K$ on $(\mathbb{R}^d, \mathcal{B}(\mathbb{R}^d))$ with invariant distribution $\pi$, and they are jointly following the $L$-lag coupling algorithm (Algorithm 1 in the main paper). As an aside, following [7] we also include the corresponding time-averaged $L$-lag estimator:

$$H_{k:m}^{(L)}(X,Y) := \frac{1}{m-k+1} \sum_{t=k}^{m} H_t^{(L)}(X,Y) \tag{2}$$

$$= \frac{1}{m-k+1} \sum_{t=k}^{m} h(X_t) + \frac{1}{m-k+1} \sum_{t=k}^{m} \sum_{j=1}^{\left\lceil \frac{\tau^{(L)} - L - t}{L} \right\rceil} h(X_{t+jL}) - h(Y_{t+(j-1)L}). \tag{3}$$

Following the proof technique for the 1-lag estimator in [7], we first prove an unbiasedness result for $H_t^{(L)}(X,Y)$. By linearity the unbiasedness of $H_{k:m}^{(L)}(X,Y)$ follows.

**Proposition 1.1.** *Under the Assumptions 2.2, 2.3 and 2.4 of the main article, $H_t^{(L)}(X,Y)$ has expectation $\mathbb{E}_{X \sim \pi}[h(X)]$, finite variance, and finite expected computing time.*

*Proof.* The proof is nearly identical to those in [13, 5, 7] and related articles, and is only reproduced here for completeness. Let $t = 0$ without loss of generality. Otherwise start the chains at $\pi_t$ rather than $\pi_0$. Secondly, we can focus on the component-wise behaviour of $H_0^{(L)}(X,Y)$ and assume $h$ takes values in $\mathbb{R}$ without loss of generality. For simplicity of notation we drop the $(L)$ superscript and write $H_0(X,Y)$ to denote $H_0^{(L)}(X,Y)$.

Define $\Delta_0 = h(X_0)$, $\Delta_j = h(X_{jL}) - h(Y_{(j-1)L})$ for $j \geq 1$, and $H_0^n(X,Y) := \sum_{j=0}^{n} \Delta_j$. By Assumption 2.3, $\mathbb{E}[\tau^{(L)}] < \infty$, so the computation time has finite expectation. When $(1+j)L \geq \tau^{(L)}$, $\Delta_j = 0$ by faithfulness (Assumption 2.4). As $\tau^{(L)} \overset{a.s.}{<} \infty$, this implies $H_0^n(X,Y) \overset{a.s.}{\to} H_0(X,Y)$ as $n \to \infty$.

We now show that $(H_0^n(X,Y))_{n\geq 0}$ is a Cauchy sequence in $L_2$, the space of random variable with finite first two moments, by showing

$$\sup_{n'\geq n} \mathbb{E}[\left(H_0^{n'}(X,Y) - H_0^n(X,Y)\right)^2] \underset{n\to\infty}{\to} 0.$$

This follows by direct calculation. Firstly by Cauchy–Schwarz,

$$\mathbb{E}[\left(H_0^{n'}(X,Y) - H_0^n(X,Y)\right)^2] = \sum_{s=n+1}^{n'}\sum_{t=n+1}^{n'} \mathbb{E}[\Delta_s\Delta_t] \leq \sum_{s=n+1}^{n'}\sum_{t=n+1}^{n'} \mathbb{E}[\Delta_s^2]^{1/2}\mathbb{E}[\Delta_t^2]^{1/2}.$$

By Hölder's inequality with $p = 1 + \eta/2$, $q = (2+\eta)/\eta$ and Assumptions 2.2 - 2.3, for any $\eta > 0$,

$$\mathbb{E}[\Delta_t^2] = \mathbb{E}[\Delta_t^2\mathbf{1}(\tau^{(L)} > (1+t)L)] \leq \mathbb{E}[\Delta_t^{2+\eta}]^{\frac{1}{1+\eta/2}}\mathbb{E}[\mathbf{1}(\tau^{(L)} > (1+t)L)]^{\frac{\eta}{2+\eta}}$$
$$< D^{\frac{1}{1+\eta/2}}(C\delta^t)^{\frac{\eta}{2+\eta}}.$$

where $\mathbb{E}[\Delta_t^{2+\eta}] \leq \mathbb{E}[M_{\mathcal{H}}(X_{tL}, Y_{(t-1)L})^{2+\eta}] \leq D$ follows from Assumptions 2.2. Overall this implies $\mathbb{E}[\left(H_0^{n'}(X,Y) - H_0^n(X,Y)\right)^2] \leq \tilde{C}\tilde{\delta}^n$ for some $\tilde{C} > 0, \tilde{\delta} \in (0,1)$ for all $n \geq 0$. Hence $(H_0^n(X,Y))_{n\geq 0}$ is a Cauchy sequence in $L_2$, and has finite first and second moments. Recall that Cauchy sequences are bounded, so we can apply the dominated convergence theorem to get,

$$\mathbb{E}[H_0(X,Y)] = \mathbb{E}[\lim_{n\to\infty} H_0^n(X,Y)] = \lim_{n\to\infty} \mathbb{E}[H_0^n(X,Y)].$$

Finally, note that by a telescoping sum argument and Assumption 2.2,

$$\lim_{n\to\infty} \mathbb{E}[H_0^n(X,Y)] = \lim_{n\to\infty} \mathbb{E}[h(X_n)] = \mathbb{E}_{X\sim P}[h(X)].$$

as required. Therefore, in general $H_t^{(L)}(X,Y)$ has expectation $\mathbb{E}_{X\sim\pi}[h(X)]$, finite variance, and a finite expected computing time. $\square$

## 1.2 Proof of Theorem 2.5

*Proof.* We consider the $L$-lag estimate in (1). Under Assumptions 2.2, 2.3 and 2.4, by Proposition 1.1 $H_t^{(L)}(X,Y)$ is an unbiased estimator of $\mathbb{E}_{X\sim\pi}[h(X)]$, for any $h \in \mathcal{H}$. Then,

$$d_{\mathcal{H}}(\pi_t, \pi) = \sup_{h\in\mathcal{H}} |\mathbb{E}_{X\sim\pi}[h(X)] - \mathbb{E}[h(X_t)]|$$

$$= \sup_{h\in\mathcal{H}} \left|\mathbb{E}\left[\sum_{j=1}^{\lceil\frac{\tau^{(L)}-L-t}{L}\rceil} h(X_{t+jL}) - h(Y_{t+(j-1)L})\right]\right|$$

$$\leq \mathbb{E}\left[\sum_{j=1}^{\lceil\frac{\tau^{(L)}-L-t}{L}\rceil} M_{\mathcal{H}}(X_{t+jL}, Y_{t+(j-1)L})\right].$$

The inequality above stems from 1) the triangle inequality applied $\lceil(\tau^{(L)} - L - t)/L\rceil$ times, and 2) the bound $|h(x) - h(y)| \leq M_{\mathcal{H}}(x,y)$ assumed in the main article. We see that increasing the lag $L$ reduces the number of applications of the triangle inequality performed above, which explains the benefits of increasing $L$. $\square$

## 1.3 Bias of Sequential Monte Carlo samplers

For an SMC sampler [4] with $N$ particles targeting $\pi$, let $(w^n, \xi^n)_{n=1}^N$ be the particle approximation of $\pi$, so that weighted averages $\sum_{n=1}^N w^n h(\xi^n)$ are consistent approximations of $\int h(x)\pi(dx)$ as $N \to \infty$ under some assumptions, e.g. [14]. We consider a particle $\xi$ drawn among $(\xi^n)_{n=1}^N$ with probabilities $(w^n)_{n=1}^N$, and we denote by $q^{(N)}$ the marginal distribution of $\xi$. Our goal is to formulate an upper bound on the total variation distance between $q^{(N)}$ and $\pi$ for fixed $N$, which is a way of studying the non-asymptotic bias of SMC samplers.

To use the proposed machinery, we embed the SMC sampler in an MCMC algorithm, following [1]. The particle independent MH (PIMH) algorithm operates as follows. Initially an SMC sampler is run, from which a particle $\xi_0$ is drawn (marginally from $q^{(N)}$), as well as a normalizing constant estimator $\hat{Z}_0$ [4]. We can think of the state of the chain as the pair $(\xi_0, \hat{Z}_0)$. At each iteration $t \geq 1$, a new SMC sampler is run and generates $(\xi^\star, \hat{Z}^\star)$. With probability $\min(1, \hat{Z}^\star/\hat{Z}_{t-1})$, the new state of the chain is set to $(\xi^\star, \hat{Z}^\star)$, otherwise it remains at $(\xi_{t-1}, \hat{Z}_{t-1})$. It is shown in [1] that this algorithm corresponds to a standard Metropolis–Hastings algorithm with independent proposals upon introducing some auxiliary variables. Therefore under some conditions, the generated chain is such that $\xi_t$ goes to $\pi$ as $t \to \infty$. We assume throughout that our three assumptions hold, which corresponds to assumptions on the performance of the SMC sampler in the present setting.

Next consider an $L$-lag coupling of such a PIMH algorithm as proposed in [9] and described in Algorithm 11. In this setting, we can characterize the distribution of the coupling time. In particular,

$$\tau^{(L)} - (L-1)\big|\hat{Z}_{L-1} \sim \text{Geometric}(\alpha(\hat{Z}_{L-1})), \tag{4}$$

where the Geometric distribution is parameterized to take integers values greater than or equal to 1, and $\alpha(\hat{Z}) := \mathbb{E}\big[\min(1, \hat{Z}^\star/\hat{Z})\big|\hat{Z}\big]$ is the acceptance probability of the PIMH chain from a state with normalizing constant estimate $\hat{Z}$. Using a monotonicity property of IMH [3], [9, Proposition 8] presents this result for 1-Lag couplings of PIMH, and (4) is a simple generalization to $L$-lag couplings; we refer to [9] for the explicit assumptions being made. Assuming that Theorem 2.5 applies, we consider the initial time $t = 0$ and obtain

$$d_{TV}(q^{(N)}, \pi) \leq \mathbb{E}\Big[\Big\lceil \frac{\tau^{(L)} - L}{L} \Big\rceil\Big]$$

$$= \mathbb{E}\Big[\mathbb{E}\Big[\Big\lceil \frac{\tau^{(L)} - (L-1) - 1}{L} \Big\rceil\Big|\hat{Z}_{L-1}\Big]\Big]$$

$$= \mathbb{E}\Big[\frac{1 - \alpha(\hat{Z}_{L-1})}{1 - (1 - \alpha(\hat{Z}_{L-1}))^L}\Big],$$

as required. Note that in the first inequality we used the fact that the total variation distance between some marginals of two multivariate distributions is less than the total variation distance between the joint distributions. The final equality follows from noting that for $G \sim \text{Geometric}(p)$ and integers $m \geq 0, n > 0$,

$$\mathbb{E}\Big[\Big\lceil \frac{G-m}{n} \Big\rceil\Big] = \sum_{k=0}^{\infty} \mathbb{P}\Big(\Big\lceil \frac{G-m}{n} \Big\rceil > k\Big) = \sum_{k=0}^{\infty} \mathbb{P}\Big(\frac{G-m}{n} > k\Big) = \frac{(1-p)^m}{1 - (1-p)^n}.$$

## 2 Couplings of MCMC algorithms

In this section, all the algorithms used in our examples are presented. These are constructions used in recent work on unbiased MCMC estimation with couplings, e.g. [7, 6, 9]. All scripts in R are available at `https://github.com/niloyb/LlagCouplings`.

We first describe algorithms to sample from maximal couplings. We then describe algorithms to sample meeting times corresponding to various couplings of MCMC algorithms.

**Maximal Couplings.** To construct $L$-lag couplings, the pair of chains needs to meet exactly whilst preserving their respective marginal distributions. This can be achieved using *maximal coupling* [8, 12], which we present below in Algorithm 1. Given variables $X \sim P, Y \sim Q$, Algorithm 1 samples jointly from $(X, Y)$ such that the marginal distributions of $X$ and $Y$ are preserved and $X$ equals $Y$ with maximal probability. It requires sampling from the distributions of $X$ and $Y$ and evaluating the ratio of their probability density functions. Below $P$ and $Q$ denote distributions of $X$ and $Y$; $p$ and $q$ denote the respective probability density functions.

For the particular case when $P = \mathcal{N}(\mu_1, \Sigma), Q = \mathcal{N}(\mu_2, \Sigma)$, we can use a *reflection-maximal coupling* [7, 2] which has deterministic computational cost. This also samples jointly from $(X, Y)$ such that the marginal distributions of $X, Y$ are preserved and $X$ equals $Y$ with maximal probability. This is given in Algorithm 2 below, where $s$ denotes the probability density function of a $d$-dimensional standard Normal. Note that in the case $\dot{Y} = \dot{X} + z$ below, we get an event $\{X = Y\}$ as required.

---
**Algorithm 1:** A maximal coupling of $P$ and $Q$
---
Sample $X \sim P$, and $W \sim \mathcal{U}(0,1)$
**if** $p(X)W \leq q(X)$ **then** set $Y = X$ and **return** $(X, Y)$
**else** sample $Y^* \sim q$ and $W^* \sim \mathcal{U}(0,1)$ until $q(Y^*)W^* > p(Y^*)$. Set $Y = Y^*$ and **return** $(X, Y)$
---

---
**Algorithm 2:** A reflection-maximal coupling of $\mathcal{N}(\mu_1, \Sigma)$ and $\mathcal{N}(\mu_2, \Sigma)$
---
Let $z = \Sigma^{-1/2}(\mu_1 - \mu_2)$ and $e = z/\|z\|$. Sample $\dot{X} \sim \mathcal{N}(0_d, \mathbf{I}_d)$, and $W \sim \mathcal{U}(0,1)$
**if** $s(\dot{X})W \leq s(\dot{X} + z)$ **then** Set $\dot{Y} = \dot{X} + z$
**else** Set $\dot{Y} = \dot{X} - 2(e^T \dot{X})e$
Set $X = \Sigma^{1/2}\dot{X} + \mu_1, Y = \Sigma^{1/2}\dot{Y} + \mu_2$, and **return** $(X, Y)$
---

When random variables $X, Y$ have discrete distributions $P = (p_1, \ldots, p_N), Q = (q_1, \ldots, q_N)$ on a finite state space, we can perform a maximal coupling with deterministic computation cost. This is given in Algorithm 3. First, we define $C = (c_1, \ldots, c_N)$ as $c_n = (p_n \wedge q_n)/S$ for $n \in \{1, \ldots, N\}$ with $S = \sum_{n=1}^{N}(p_n \wedge q_n)$. The notation $a \wedge b$ stands for the minimum of $a$ and $b$. We then define $P'$ and $Q'$ as $p_n' = (p_n - p_n \wedge q_n)/(1 - S)$, and $q_n' = (q_n - p_n \wedge q_n)/(1 - S)$. These $P'$ and $Q'$ are probability vectors and computing them takes $\mathcal{O}(N)$ operations. Note that the total variation distance between $P$ and $Q$ is equal to $1 - S$, and that $P'$ and $Q'$ have disjoint supports.

---
**Algorithm 3:** A maximal coupling of $P = (p_1, \ldots, p_N), Q = (q_1, \ldots, q_N)$
---
Sample $U \sim \mathcal{U}(0,1)$
**if** $U < S$ **then** Sample $X$ from $C$, define $Y = X$ and **return** $(X, Y)$
**else** Sample $X$ from $P'$, $Y$ from $Q'$ independently, and **return** $(X, Y)$
---

## 2.1 Random walk Metropolis–Hastings

We couple a pair of random walk Metropolis–Hastings chains in Sections 2.2.1 and 2.2.2 using Algorithm 4 with step sizes $\sigma_{\mathrm{MH}} = 0.5$ and $\sigma_{\mathrm{MH}} = 1$ respectively. We could also modify the algorithm to use more general proposal kernels $q(\cdot, \cdot)$, provided that we can sample from a maximal coupling of $q(x, \cdot)$ and $q(y, \cdot)$ for any pair $x, y$.

---
**Algorithm 4:** Gaussian random walk Metropolis–Hastings
---
**Input:** lag $L \geq 1$, random walk step size $\sigma_{\mathrm{MH}}$
**Output:** meeting time $\tau^{(L)}$; chains $(X_t)_{0 \leq t \leq \tau^{(L)}}, (Y_t)_{0 \leq t \leq \tau^{(L)} - L}$
Initialize: generate $X_0 \sim \pi_0$ and $Y_0 \sim \pi_0$
**for** $t = 1, \ldots, L$ **do**
    Sample proposal $X^* \sim \mathcal{N}(X_{t-1}, \sigma_{\mathrm{MH}}^2)$
    Sample $U \sim \mathcal{U}(0,1)$
    **if** $U \leq \frac{\pi(X^*)}{\pi(X_{t-1})}$, **then** set $X_t = X^*$ ; **else** set $X_t = X_{t-1}$
**end**
**for** $t > L$ **do**
    Sample proposals $X^* \sim \mathcal{N}(X_{t-1}, \sigma_{\mathrm{MH}}^2), Y^* \sim \mathcal{N}(Y_{t-1-L}, \sigma_{\mathrm{MH}}^2)$ jointly using maximal (or reflection-maximal) coupling
    Sample $U \sim \mathcal{U}(0,1)$
    **if** $U \leq \frac{\pi(X^*)}{\pi(X_{t-1})}$, **then** set $X_t = X^*$ ; **else** set $X_t = X_{t-1}$
    **if** $U \leq \frac{\pi(Y^*)}{\pi(Y_{t-1-L})}$, **then** set $Y_{t-L} = Y^*$ ; **else** set $Y_{t-L} = Y_{t-1-L}$
    **if** $X_t = Y_{t-L}$ **then** **return** $\tau^{(L)} := t$, and the chains $(X_t)_{0 \leq t \leq \tau^{(L)}}, (Y_t)_{0 \leq t \leq \tau^{(L)} - L}$
**end**
---

## 2.2 MCMC algorithms for the Ising model

**Single site Gibbs (SSG).** Our implementation of single site Gibbs (SSG) scans all the sites of the lattice systematically. We recall that the full conditionals of the Gibbs sampling updates are Bernoulli distributed; we denote by $p(\beta, X_{-i})$ the conditional probability of site $X_i$ being equal to $+1$ given the other sites. The algorithm to sample meeting times is given in Algorithm 5. The SSG results in Section 3.1 are generated using Algorithm 5 with $\beta = 0.46$.

---

**Algorithm 5:** Single Site Gibbs sampler for the Ising model

---

**Input:** lag $L \geq 1$, and inverse temperature $\beta$
**Output:** meeting time $\tau^{(L)}$; chains $(X_t)_{0 \leq t \leq \tau^{(L)}}, (Y_t)_{0 \leq t \leq \tau^{(L)} - L}$
Initialize: generate $X_0 \sim \pi_0$ and $Y_0 \sim \pi_0$
**for** $t = 1, \ldots, L$ **do**
    **for** *site* $i = 1, \ldots, 32 \times 32$ **do**
        Sample $X_{i,t}|X_{-i,t} \sim \text{Bernoulli}(p(\beta, X_{-i,t}))$
    **end**
**end**
**for** $t > L$ **do**
    **for** *site* $i = 1, \ldots, 32 \times 32$ **do**
        Sample $X_{i,t}|X_{-i,t} \sim \text{Bernoulli}(p(\beta, X_{-i,t}))$ and
        $Y_{i,t-L}|Y_{-i,t-L} \sim \text{Bernoulli}(p(\beta, Y_{-i,t-L}))$ jointly using e.g. Algorithm 3
    **end**
    **if** $X_t = Y_{t-L}$ **then return** $\tau^{(L)} := t$, and the chains $(X_t)_{0 \leq t \leq \tau^{(L)}}, (Y_t)_{0 \leq t \leq \tau^{(L)} - L}$
**end**

---

**Parallel tempering (PT).** For parallel tempering, we introduce $C$ chains denoted by $x^{(1)}, \ldots, x^{(C)}$. Each chain $X^{(c)}$ targets the distribution $\pi_{\beta^{(c)}}$ where $(\beta^{(c)})_{c=1}^C$ are positive values interpreted as inverse temperatures. In the example in Section 3.1, we have $C = 12$, $\beta^{(1)} = 0.3$, $\beta^{(C)} = 0.46$, and the intermediate $\beta^{(c)}$ are equispaced. The frequency of proposed swap moves is denoted by $\omega$ and set to 0.02. This is in no way optimal, see [11] for practical tuning strategies. Our implementation of a coupled PT algorithm is given below in Algorithm 6.

Note that in the case of parallel tempering, meetings occur when all the $C$ pairs of chains have met. This incurs a trade-off: increasing the number of chains might improve the performance of the marginal algorithm but could also complicate the occurrence of meetings; see [11] for other trade-offs associated with the number of chains in parallel tempering.

## 2.3 Pólya-Gamma Gibbs sampler

Algorithm 7 couples the Pólya-Gamma sampler for Bayesian logistic regression [10], as in Section 3.2 with prior $\mathcal{N}(b, B)$ on $\beta$ for $b = 0$, $B = 10I_d$. Parameters $\beta, \tilde{\beta} \in \mathbb{R}^d$, $W, \tilde{W} \in \mathbb{R}_+^n$ correspond to the vectors of regression coefficients and auxiliary variables respectively for the pair of chains. The vector $\tilde{y}$ is defined as $\tilde{y} = y - 1/2$, where $y$ is the vector of responses $y \in \{0, 1\}^n$.

In the algorithm, $PG(1, c)$ refers to the Pólya-Gamma variable in the notation of [10]. The notation $X|rest$ refers to the conditional distribution of $X$ given all the other variables. The tilde notation refers to components of the second chain. The coupling here was also used in [7].

## 2.4 Hamiltonian Monte Carlo

Algorithm 8 couples Hamiltonian Monte Carlo (HMC), as used in Section 3.2. We follow the coupling construction from [6]; see also references therein. For simplified notation, we will use $K_p(\beta, \cdot \; ; \; \epsilon_{\text{HMC}}, S_{\text{HMC}})$ to denote the leapfrog integration and the accept-reject part of HMC from position $\beta \in \mathbb{R}^d$ with momentum $p \in \mathbb{R}^d$. Here $\epsilon_{\text{HMC}}$ and $S_{\text{HMC}}$ correspond to the step size and the number of steps respectively in the leapfrog integration scheme.

**Algorithm 6:** Parallel tempering for the Ising model

---

**Input:** lag $L \geq 1$, and inverse temperatures $(\beta^{(c)})_{c=1}^{C}$

**Output:** meeting time $\tau^{(L)}$, chains $(X_t^{(c)})_{0 \leq t \leq \tau^{(L)}}, (Y_t^{(c)})_{0 \leq t \leq \tau^{(L)} - L}$ for $c = 1, \ldots, C$

Initialize: generate $X_0^{(c)} \sim \pi_0$ and $Y_0^{(c)} \sim \pi_0$ for each chain $c = 1, \ldots, C$

**for** $t = 1, \ldots, L$ **do**

    Sample $U \sim \mathcal{U}(0,1)$

    **if** $U < \omega$ **then**

        Define $X_t^{(c)} = X_{t-1}^{(c)}$ for all $c = 1, \ldots, C$

        **for** $c = 1, \ldots, C-1$ **do**

            Swap chain states $X_t^{(c)}, X_t^{(c+1)}$ with probability $\min\left(1, \frac{\pi_{\beta^{(c)}}(X_t^{(c+1)})\pi_{\beta^{(c+1)}}(X_t^{(c)})}{\pi_{\beta^{(c)}}(X_t^{(c)})\pi_{\beta^{(c+1)}}(X_t^{(c+1)})}\right)$

        **end**

    **else**

        **for** $c = 1, \ldots, C$ **do**

            Update $X_t^{(c)} \sim SSG(X_{t-1}^{(c)}; \beta^{(c)})$

        **end**

    **end**

**end**

**for** $t > L$ **do**

    Sample $U \sim \mathcal{U}(0,1)$

    **if** $U < \omega$ **then**

        Define $X_t^{(c)} = X_{t-1}^{(c)}$ and $Y_{t-L}^{(c)} = Y_{t-L-1}^{(c)}$ for all $c$

        **for** $c = 1, \ldots, C-1$ **do**

            Sample $U^{(c)} \sim \mathcal{U}(0,1)$

            **if** $U^{(c)} \leq \frac{\pi_{\beta^{(c)}}(X_t^{(c+1)})\pi_{\beta^{(c+1)}}(X_t^{(c)})}{\pi_{\beta^{(c)}}(X_t^{(c)})\pi_{\beta^{(c+1)}}(X_t^{(c+1)})}$, swap chain states $X_t^{(c)}, X_t^{(c+1)}$

            **if** $U^{(c)} \leq \frac{\pi_{\beta^{(c)}}(Y_{t-L}^{(c+1)})\pi_{\beta^{(c+1)}}(Y_{t-L}^{(c)})}{\pi_{\beta^{(c)}}(Y_{t-L}^{(c)})\pi_{\beta^{(c+1)}}(X_{t-L}^{(c+1)})}$, swap chain states $Y_{t-L}^{(c)}, Y_{t-L}^{(c+1)}$

        **end**

    **else**

        **for** $c = 1, \ldots, C$ **do**

            Update $X_t^{(c)} \sim SSG(X_{t-1}^{(c)}; \beta^{(c)})$ and $Y_{t-L}^{(c)} \sim SSG(Y_{t-L-1}^{(c)}; \beta^{(c)})$

            jointly using coupled SSG (see Algorithm 5)

        **end**

    **end**

    **if** $X_t^{(c)} = Y_{t-L}^{(c)}$ *for* $c = 1, \ldots, C$ **then**

        **return** $\tau^{(L)} := t$, and the chains $(X_t^{(c)})_{0 \leq t \leq \tau^{(L)}}, (Y_t^{(c)})_{0 \leq t \leq \tau^{(L)} - L}$ for all $c$.

    **end**

**end**

---

We write $\bar{K}_{\mathrm{RWMH}}((\beta, \tilde{\beta}), \cdot \ ; \sigma_{\mathrm{MH}})$ to denote the kernel of the coupled random walk Metropolis–Hastings algorithm (Algorithm 4) with step size $\sigma_{\mathrm{MH}}$. Mixture parameter $\gamma$ corresponds to the probability of selecting kernel $\bar{K}_{\mathrm{RWMH}}((\beta, \tilde{\beta}), \cdot \ ; \sigma_{\mathrm{MH}})$ from a mixture of the kernels $K_p(\beta, \cdot \ ; \epsilon_{\mathrm{HMC}}, S_{\mathrm{HMC}})$ and $\bar{K}_{\mathrm{RWMH}}((\beta, \tilde{\beta}), \cdot \ ; \sigma_{\mathrm{MH}})$. The HMC results in Section 3.2 are generated using Algorithm 8 with $\epsilon_{\mathrm{HMC}} = 0.025$, $S_{\mathrm{HMC}} = 4, 5, 6, 7$, $\gamma = 0.05$ and $\sigma_{\mathrm{MH}} = 0.001$.

Note that reflection-maximal coupling can also be used to draw the momenta in coupled Hamiltonian Monte Carlo, as discussed in [2, 6].

---

**Algorithm 7:** Pólya-Gamma Gibbs Coupling

---

**Input:** lag $L \geq 1$, response $y \in \{0,1\}^n$ and design matrix $X \in \mathbb{R}^{n \times d}$

**Output:** meeting time $\tau^{(L)}$; chains $(\beta_t)_{0 \leq t \leq \tau^{(L)}}, (\tilde{\beta}_t)_{0 \leq t \leq \tau^{(L)} - L}$

Initialize: generate $\beta_0 \sim \pi_0$ and $\tilde{\beta}_0 \sim \pi_0$

**for** $t = 1, \ldots, L$ **do**
> Sample $W_{t,i}|rest \sim PG(1, |x_i^T \beta_{t-1}|)$ for $i = 1, \ldots, n$
> Sample $\beta_t|rest \sim \mathcal{N}(\Sigma(W_t)(X^T \tilde{y} + B^{-1}b), \Sigma(W_t))$ for
> $\Sigma(W_t) = (X^T \mathrm{diag}(W_t)X + B^{-1})^{-1}$

**end**

**for** $t > L$ **do**
> Sample $W_{t,i}|rest$ and $\tilde{W}_{t-L,i}|r\tilde{e}st$, jointly using maximal couplings of $PG(1, |x_i^T \beta_{t-1}|)$ and
> $PG(1, |x_i^T \tilde{\beta}_{t-L-1}|)$, for $i = 1, \ldots, n$, by noting that the ratio of density functions of two
> Pólya-Gamma random variables is tractable:
>
> $$\forall x > 0, \; \frac{PG(x; 1, c_1)}{PG(x; 1, c_2)} = \frac{\cosh(c_2/2)}{\cosh(c_1/2)} \exp\left( - \left( \frac{c_2^2}{2} - \frac{c_1^2}{2} \right)x \right)$$
>
> Sample $\beta_t|rest$ and $\tilde{\beta}_{t-L}|r\tilde{e}st$ from a maximal coupling of
>
> $$\mathcal{N}(\Sigma(W_t)(X^T\tilde{y} + B^{-1}b), \Sigma(W_t)) \text{ and } \mathcal{N}(\Sigma(\tilde{W}_{t-L})(X^T\tilde{y} + B^{-1}b), \Sigma(\tilde{W}_{t-L}))$$
>
> **if** $\beta_t = \tilde{\beta}_{t-L}$ **then return** $\tau^{(L)} := t$, and the chains $(\beta_t)_{0 \leq t \leq \tau^{(L)}}, (\tilde{\beta}_t)_{0 \leq t \leq \tau^{(L)} - L}$.

**end**

---

---

**Algorithm 8:** Hamiltonian Monte Carlo

---

**Input:** lag $L \geq 1$, mixture parameter $\gamma \in (0, 1)$, and random walk step size $\sigma_{\mathrm{MH}}$

**Output:** meeting time $\tau^{(L)}$; chains $(\beta_t)_{0 \leq t \leq \tau^{(L)}}, (\tilde{\beta}_t)_{0 \leq t \leq \tau^{(L)} - L}$

Initialize: generate $\beta_0 \sim \pi_0$ and $\tilde{\beta}_0 \sim \pi_0$

**for** $t = 1, \ldots, L$ **do**
> Sample momentum $p^* \sim \mathcal{N}(0_d, \mathbf{I}_d)$ and sample $\beta_t \sim K_{p^*}(\beta_{t-1}, \cdot \, ; \, \epsilon_{\mathrm{HMC}}, S_{\mathrm{HMC}})$

**end**

**for** $t > L$ **do**
> Sample $U \sim \mathcal{U}(0, 1)$
> **if** $U \leq \gamma$ **then**
> > Sample $\beta_t, \tilde{\beta}_{t-L} \sim \bar{K}_{\mathrm{RWMH}}((\beta_{t-1}, \tilde{\beta}_{t-L-1}), \cdot \, ; \, \sigma_{\mathrm{MH}})$ using Algorithm 4
>
> **else**
> > Sample common momentum $p^* \sim \mathcal{N}(0_d, \mathbf{I}_d)$
> > Sample $\beta_t \sim K_{p^*}(\beta_{t-1}, \cdot \, ; \, \epsilon_{\mathrm{HMC}}, S_{\mathrm{HMC}})$ and $\tilde{\beta}_{t-L} \sim K_{p^*}(\tilde{\beta}_{t-1-L}, \cdot \, ; \, \epsilon_{\mathrm{HMC}}, S_{\mathrm{HMC}})$
>
> **end**
>
> **if** $\beta_t = \tilde{\beta}_{t-L}$ **then return** $\tau^{(L)} := t$, and the chains $(\beta_t, W_t)_{0 \leq t \leq \tau^{(L)}}, (\tilde{\beta}_t, \tilde{W}_t)_{0 \leq t \leq \tau^{(L)} - L}$

**end**

---

## 2.5 Metropolis–adjusted Langevin Algorithm

The Metropolis–adjusted Langevin Algorithm (MALA) can be coupled as in random walk Metropolis–Hastings, as it corresponds to a particular choice of proposal distribution. For simplicity of notation we use $q_\sigma(X, \cdot) \sim \mathcal{N}(X + \frac{1}{2}\sigma^2 \nabla \log \pi(X), \sigma^2 \mathbf{I}_d)$ to denote the Langevin proposal. The MALA results in Section 3.3 are generated using Algorithm 9 with $\sigma = d^{-1/6}$ for $d = 50, 100, 200, 300, 400, 500, 600, 800, 1000$.

---

**Algorithm 9:** MALA

---

**Input:** lag $L \geq 1$, random walk step size $\sigma$
**Output:** meeting time $\tau^{(L)}$; chains $(X_t)_{0 \leq t \leq \tau^{(L)}}, (Y_t)_{0 \leq t \leq \tau^{(L)} - L}$
Initialize: generate $X_0 \sim \pi_0$ and $Y_0 \sim \pi_0$
**for** $t = 1, \ldots, L$ **do**
    Sample proposal $X^* \sim q_\sigma(X_{t-1}, \cdot)$
    Sample $U \sim \mathcal{U}(0,1)$
    **if** $U \leq \frac{\pi(X^*)q_\sigma(X^*,X_{t-1})}{\pi(X_{t-1})q_\sigma(X_{t-1},X^*)}$, **then** set $X_t = X^*$ ; **else** set $X_t = X_{t-1}$
**end**
**for** $t > L$ **do**
    Sample proposals $X^* \sim q_\sigma(X_{t-1}, \cdot), Y^* \sim q_\sigma(Y_{t-1-L}, \cdot)$ jointly via reflection-maximal
    coupling of Algorithm 2
    Sample $U \sim \mathcal{U}(0,1)$
    **if** $U \leq \frac{\pi(X^*)q_\sigma(X^*,X_{t-1})}{\pi(X_{t-1})q_\sigma(X_{t-1},X^*)}$, **then** set $X_t = X^*$ ; **else** set $X_t = X_{t-1}$
    **if** $U \leq \frac{\pi(Y^*)q_\sigma(Y^*,Y_{t-1-L})}{\pi(Y_{t-1-L})q_\sigma(Y_{t-1-L},Y^*)}$, **then** set $Y_{t-L} = Y^*$ ; **else** set $Y_{t-L} = Y_{t-1-L}$

    **if** $X_t = Y_{t-L}$ **then return** $\tau^{(L)} := t$, and the chains $(X_t)_{0 \leq t \leq \tau^{(L)}}, (Y_t)_{0 \leq t \leq \tau^{(L)} - L}$
**end**

---

## 2.6 Unadjusted Langevin Algorithm

Unadjusted Langevin proceeds as MALA but without the MH acceptance step. Thus an algorithm to sample meeting times for coupled ULA chains follows from the algorithm described for coupled MALA algorithm, simply by removing the acceptance steps. As before, we use $q_\sigma(X, \cdot) \sim \mathcal{N}(X + \frac{1}{2}\sigma^2 \nabla \log \pi(X), \sigma^2 \mathbf{I}_d)$ to denote the Langevin proposal. The ULA results in Section 3.3 are generated using Algorithm 10 with $\sigma = 0.1 d^{-1/6}$ for $d = 50, 100, 200, 300, 400, 500, 600, 800, 1000$.

---

**Algorithm 10:** ULA

---

**Input:** lag $L \geq 1$, random walk step size $\sigma$
**Output:** meeting time $\tau^{(L)}$; chains $(X_t)_{0 \leq t \leq \tau^{(L)}}, (Y_t)_{0 \leq t \leq \tau^{(L)} - L}$
Initialize: generate $X_0 \sim \pi_0$ and $Y_0 \sim \pi_0$
**for** $t = 1, \ldots, L$ **do**
    Sample $X_t \sim q_\sigma(X_{t-1}, \cdot)$
**end**
**for** $t > L$ **do**
    Sample $X_t \sim q_\sigma(X_{t-1}, \cdot), Y_{t-L} \sim q_\sigma(Y_{t-1-L}, \cdot)$ jointly via reflection-maximal coupling of
    Algorithm 2

    **if** $X_t = Y_{t-L}$ **then return** $\tau^{(L)} := t$, and the chains $(X_t)_{0 \leq t \leq \tau^{(L)}}, (Y_t)_{0 \leq t \leq \tau^{(L)} - L}$
**end**

---

## 2.7 Particle independent Metropolis–Hastings

By construction, $\tau^{(L)} > L$ almost surely for all the above couplings. Here we describe a version of coupled particle independent Metropolis–Hastings (PIMH) which allows coupling at the first step, such that $\tau^{(L)} = L$ can occur with positive probability. This coupling was introduced in [9].

---
**Algorithm 11:** Particle independent Metropolis–Hastings
---
**Input:** lag $L \geq 1$, and SMC sampler targeting $\pi$

**Output:** meeting time $\tau^{(L)}$; chains $(\xi_t, Z_t)_{0 \leq t \leq \tau^{(L)}}$, $(\tilde{\xi}_t, \tilde{Z}_t)_{0 \leq t \leq \tau^{(L)} - L}$ where $Z_t, \tilde{Z}_t$ are unbiased estimates of the normalizing constant of $\pi$

Initialize: Sample $\xi_0, Z_0$ from the SMC sampler

**for** $t = 1, \ldots, (L-1)$ **do**
    Sample proposal $\xi^*, Z^*$ from the SMC sampler
    Sample $U \sim \mathcal{U}(0, 1)$
    **if** $U \leq \frac{Z^*}{Z_{t-1}}$, **then** set $\xi_t = \xi^*, Z_t = Z^*$ ; **else** set $\xi_t = \xi_{t-1}, Z_t = Z_{t-1}$
**end**

**for** $t = L$ **do**
    Sample proposal $\xi^*, Z^*$ from the SMC sampler
    Sample $U \sim \mathcal{U}(0, 1)$
    **if** $U \leq \frac{Z^*}{Z_{L-1}}$, **then** set $\xi_L = \xi^*, Z_L = Z^*$ ; **else** set $\xi_L = \xi_{L-1}, Z_L = Z_{L-1}$
    Set $\tilde{\xi}_0 = \xi^*, \tilde{Z}_0 = Z^*$
**end**

**for** $t > L$ **do**
    Sample proposal $\xi^*, Z^*$ from the SMC sampler
    Sample $U \sim \mathcal{U}(0, 1)$
    **if** $U \leq \frac{Z^*}{Z_{t-1}}$, **then** set $\xi_t = \xi^*, Z_t = Z^*$ ; **else** set $\xi_t = \xi_{t-1}, Z_t = Z_{t-1}$
    **if** $U \leq \frac{Z^*}{\tilde{Z}_{t-L-1}}$, **then** set $\tilde{\xi}_{t-L} = \xi^*, \tilde{Z}_{t-L} = Z^*$ ; **else** set $\tilde{\xi}_{t-L} = \tilde{\xi}_{t-L-1}, \tilde{Z}_{t-L} = \tilde{Z}_{t-L-1}$
    **if** $\xi_t = \tilde{\xi}_{t-L}, Z_t = \tilde{Z}_{t-L}$ **then**
        **return** $\tau^{(L)} := t$, and the chains $(\xi_t, Z_t)_{0 \leq t \leq \tau^{(L)}}$, $(\tilde{\xi}_t, \tilde{Z}_t)_{0 \leq t \leq \tau^{(L)} - L}$.
    **end**
**end**
---