[Reviews · NeurIPS 2019]

Reviewer 1



The authors generalize 1-lag coupling of the chains to L-lag coupling and provide upper bounds on some distribution distances including the total variation and 1-Wasserstein distance. This bound serves as a convergence check for MCMC, e.g., to stop the burn-in phase. The main contributions of the paper are 1) deriving a computable bound of the distribution distance between two (L-lagged) chains, and 2) presenting algorithms (e.g., Coupled Random-Walk Metropolis-Hastings, Coupled HMC, etc.) using the bound as a stopping criterion for burn-in. Unfortunately, the second part together with the proof of the bound is in the supplementary material. The presented bound and method to compute it is, to the best of knowledge, novel and significantly extends the state-of-the-art. My only concern is about the presentation/composition of the paper: The bound requires to compute the (L-lag) meeting time. The algorithm to compute it compares samples (with time lag L) of different chains and stops when they coincidence. Naively, this will never happen for continuous values; hence, the algorithm assumes some coupled/joint MCMC kernel. Some viable variants are presented in the appendix. While those are based on prior work, I would still expect them to be part of the main body. Without the supplementary material, I would not consider the presented work as self-containing and reproducible. In addition, I would expect a broader discussion of relevant literature, also in the context of the applications of the bound, such as detecting convergence. For example, there are methods to quantify bias in a sample (e.g., "A Kernelized Stein Discrepancy for Goodness-of-fit Tests", Liu et al) or two-sample tests to measure sample discrepancy (e.g., "A Kernel Two-Sample Test", Gretton et al). Both approaches may also be used to detect MCMC convergence and could be considered as baselines in the experiments. After reading the authors' feedback, I revised my vote from 6 to 7. They addressed my concerns and suggested some changes so that, overall, I'm inclined accepting the paper.

Reviewer 2



The paper reads well, the content is very clearly presented. The stylized example and the application section provide a convincing illustration. However, the paper is not very original, not from a mathematical perspective at least. A reader may understand that the motivation for using a L-lag coupling rather than a 1-lag coupling, is to obtain sharper bounds'' [line 40]. First, it could be useful to provide some intuition behind this fact. Second, the authors should try to compare the convergence speed of the L-lag bound estimator with the corresponding 1-lag bound estimator. The examples show that the L-lag bound estimators are typically more stable but a quantitative analysis appears compelling. It would thus help a practitioner to know if the L-lag bound worths being used since for a replication of the L-lag coupling is typically much more computationally involved than an equivalent 1-lag replica. In the Application section, the Authors should mention if the assumptions 2.2--2.3 are satisfied. Minor comment l106 N times independently

Reviewer 3



Strength - A crisply-written paper, targeting an important problem: assessing the convergence of MCMC methods. - The proposed algorithm provides practical bounds on the distance between the marginal distribution of Markov chains and the target distribution. Estimating such bounds is a challenging research problem in computational statistics, and this paper proposes a useful method based on L-Lag couplings. - Empirical evaluation verifies the theoretical claims and demonstrates the practicality of the proposed method on several common sampling algorithms. - A fair assessment of the proposed method, with experiments demonstrating both efficacy and the failure case (sec 2.2.2), It is worth pointing out the failure case should not be considered as a weakness of this paper as it is beneficial for understanding the proposed method. Weakness - Convergence estimates from the proposed method could be incorrect when the underlying MCMC sampler does not mix well, e.g. in high-dimensional and/or multimodal target distribution settings. This issue is pointed out by the authors through numerical experiments and deserves more investigation.

[Author Response · NeurIPS 2019]

We thank the reviewers for their thorough comments and feedback.

Reviewer #1: *The first and main concern is that, without the supplementary material, the article is not self-contained nor reproducible.*

We propose to modify the article as follows to address this concern.

One concrete example of coupling of MCMC algorithm will be moved to the main text. These couplings build on previous work but indeed are not widely known. Thus we will add a full description of a coupled kernel in Section 2.2.1.

We also propose to add some motivation and explanation (five to ten lines) for the proposed bounds, just after Theorem 2.5 in the main text. The supplementary material will still contain the full formal proof.

*A secondary concern of the reviewer is a lack of discussion of some relevant articles on kernel methods.*

We thank the reviewer for pointing out references which we will add. The articles mentioned provide sample quality assessments but are not directly approximating the TV or the Wasserstein distance between $\pi_t$ and $\pi$. Thus, we are not aware of articles where these types of assessment are used to choose a burn-in value, or to obtain plots similar to the ones we show; our understanding is that these tools have been used to assess the approximation obtained post burn-in, or compare the bias of asymptotically biased samplers.

We propose to add some motivation for considering the TV and 1-Wasserstein distances, in Section 2, as these distances are practically useful: the TV controls the error made in approximating probability masses under the target (e.g. when plotting histograms of the target marginals, or when calculating credible intervals of posterior distributions), while the 1-Wasserstein controls all first moments. Further, TV and Wasserstein are used in most theoretical studies, which allows for comparison between our proposed bounds and established results, as we illustrate in Section 3.3.

Among other relevant papers we will also add a reference to "A simulation approach to convergence rates for Markov chain Monte Carlo algorithms" by Rosenthal & Cowles.

We will follow the reviewer's insightful suggestions on how to improve the presentation of the article and remove some redundancies. In passing $a \vee b$ denoted the maximum between $a$ and $b$, which we can denote by $\max(a, b)$ instead.

Reviewer #2: *The first criticism is about the originality of the proposed bounds.*

We agree with the reviewer that the introduction of a lag of $L$ instead of a lag of 1 as in reference [16] looks incremental from a mathematical point or view. However it is key to the obtention of practical bounds.

From the methodological point of view, the contribution is original. The idea of the upper bound on the TV was suggested in the last paragraphs of [16], but not implemented or discussed anywhere, as far as we know. The proposed method resembles Johnson's diagnostics, which we discuss in details, but it seems much more generically applicable.

*The second and main concern is about the choice of L, its impact on the bounds, and its cost.*

We propose to elaborate the discussion on the choice of $L$ and the appeal of using $L > 1$. We can add an explicit discussion of the cost of obtaining $\tau^{(L)}$ as a function of $L$, and how the TV upper bounds becomes less vacuous for all $t$ as $L \to \infty$. Overall we do not have strong theoretical results to guide the choice of $L$, and hope that the article will motivate further research on the topic. In the meantime, Figure 2 and, mostly, Figure 3 show the practical benefit of choosing $L > 1$.

*A final concern is about discussing the verification of assumptions in the context of the applications.* We will add comments on this but will mostly refer to [16] where such assumptions are discussed in details.

Reviewer #3:

We thank the reviewer for their supporting words regarding our efforts to point out the weaknesses of the proposed method. Regarding software implementation, designing coupling schemes for any given MCMC algorithm can require some work, nonetheless this has been done already for several popular algorithms, e.g. Hamiltonian Monte Carlo. These couplings could indeed be integrated into some software such as PyMC3, as suggested by the reviewer. We can add some discussion on this in the discussion.

[Meta-Review · NeurIPS 2019]

After discussion, all agree that this paper makes a significant contribution and merits acceptance. These results on estimating MCMC convergence with L-lag couplings will be of broad interest to the NeurIPS community. Please take the reviewers' constructive feedback into account and follow through on your promises to improve the paper as stated in the rebuttal.